# COVID-19 Vaccines: Fear of Side Effects among German Health Care Workers

**DOI:** 10.3390/vaccines10050689

**Published:** 2022-04-28

**Authors:** Christopher Holzmann-Littig, Tamara Frank, Christoph Schmaderer, Matthias C. Braunisch, Lutz Renders, Peter Kranke, Maria Popp, Christian Seeber, Falk Fichtner, Bianca Littig, Javier Carbajo-Lozoya, Joerg J. Meerpohl, Bernhard Haller, Christine Allwang

**Affiliations:** 1Department of Nephrology, School of Medicine, Technical University of Munich, Klinikum Rechts der Isar, 81675 Munich, Germany; christoph.schmaderer@mri.tum.de (C.S.); matthias.braunisch@mri.tum.de (M.C.B.); lutz.renders@tum.de (L.R.); bianca.littig@web.de (B.L.); javier.carbajolozoya@mri.tum.de (J.C.-L.); 2TUM Medical Education Center, School of Medicine, Technical University of Munich, 81675 Munich, Germany; 3Department of Psychosomatic Medicine and Psychotherapy, School of Medicine, Technical University of Munich, Klinikum Rechts der Isar, 81675 Munich, Germany; tamara.frank@mri.tum.de (T.F.); christine.allwang@mri.tum.de (C.A.); 4Department of Anesthesiology, Intensive Care, Emergency Medicine and Pain Medicine, University Hospital Wuerzburg, 97080 Wuerzburg, Germany; kranke_p@ukw.de (P.K.); popp_m4@ukw.de (M.P.); 5Department of Anesthesiology and Intensive Care, University Hospital Leipzig, 04103 Leipzig, Germany; christian.seeber@medizin.uni-leipzig.de (C.S.); falk.fichtner@medizin.uni-leipzig.de (F.F.); 6Medical Center & Faculty of Medicine, Institute for Evidence in Medicine, University of Freiburg, 79110 Freiburg, Germany; meerpohl@ifem.uni-freiburg.de; 7Cochrane Germany, Cochrane Germany Foundation, 79110 Freiburg, Germany; 8Institute of AI and Informatics in Medicine, School of Medicine, Technical University of Munich, Klinikum Rechts der Isar, 81675 Munich, Germany; bernhard.haller@mri.tum.de

**Keywords:** COVID-19, vaccine hesitancy, health care workers, side-effects, fears

## Abstract

(1) Background: Health care workers (HCWs) play a key role in increasing anti-COVID vaccination rates. Fear of potential side effects is one of the main reasons for vaccine hesitancy. We investigated which side effects are of concern to HCWs and how these are associated with vaccine hesitancy. (2) Methods: Data were collected in an online survey in February 2021 among HCWs from across Germany with 4500 included participants. Free-text comments on previously experienced vaccination side effects, and fear of short- and long-term side effects of the COVID-19 vaccination were categorized and analyzed. (3) Results: Most feared short-term side effects were vaccination reactions, allergic reactions, and limitations in daily life. Most feared long-term side effects were (auto-) immune reactions, neurological side effects, and currently unknown long-term consequences. Concerns about serious vaccination side effects were associated with vaccination refusal. There was a clear association between refusal of COVID-19 vaccination in one’s personal environment and fear of side effects. (4) Conclusions: Transparent information about vaccine side effects is needed, especially for HCW. Especially when the participants’ acquaintances advised against vaccination, they were significantly more likely to fear side effects. Thus, further education of HCW is necessary to achieve good information transfer in clusters as well.

## 1. Introduction

The coronavirus disease 2019 (COVID-19) pandemic has caused and still causes impairment and detriment on health care systems and economies, but foremost, on the lives of millions of people worldwide. Besides the quickly initiated hygienic and behavioral measures, the development of effective vaccines targeting the virus had prevailing priority. Since uncontrolled infection causes an enormous burden on national health care systems [1], increased mortality, and possibly causes long-term health consequences [2], the possibility to vaccinate presents the most efficient solution in controlling and ending the pandemic. Vaccination programs can lead to herd immunity without demanding the majority of a population getting infected and without stressing health care systems [3]. However, this requires a sufficient proportion of a country’s population to be vaccinated. Recent studies indicate that due to new virus variants, up to 90% of the population need to be vaccinated against COVID-19 in order to achieve herd immunity [4].

Thus, the effectiveness of vaccination programs depends on the individuals’ willingness to be vaccinated. Doubts and worries can negatively influence this willingness and lead to vaccine hesitancy or even rejection [5]. Vaccine hesitancy [6] has been identified by the World Health Organization as one of the 10 threats to global health even before the outbreak of COVID-19 [7].

Health care workers (HCWs) play a key role within vaccination programs and their willingness to get vaccinated is therefore of particular importance. Due to their interaction with patients, HCWs might be more exposed to COVID-19 and play a central role in nosocomial transmission [8]. At the same time, their health and ability to work is essential during a pandemic to keep health care systems stable and running. HCWs also function as multipliers, role models, influencers, and trusted advisors for their patients, and also for their families and friends. Their doctor’s recommendation is an important factor for people when considering whether or not to get the COVID-19 vaccine [9], and can even function as a predictor of vaccine enthusiasm [10]. In Germany, according to a study by the Robert Koch Institute, the vaccination rate of German hospital staff was 83% (primary vaccination) in April 2021, with significant differences between physicians and non-physicians [11]. Our study had already suggested a COVID-19 vaccination acceptance rate of 91.7% of health care professionals in Germany in 2021 [12]. This is mirrored, at least, for German hospital staff according to a recent study by the Robert Koch Institute, with a rate of 92% fully vaccinated study participants [13]. However, there is still a proportion of unvaccinated health care workers in Germany. Providing credible and reliable information regarding the risks and benefits of vaccines is therefore a central responsibility of HCWs in supporting patients in their informed decision making [7].

By the end of 2020 and early 2021, the first COVID-19 vaccines were approved and available in Germany, and HCWs were prioritized and among the first to receive them [14].

However, many HCWs appear to be hesitant about receiving the COVID-19 vaccine [15,16,17]. Several demographic factors such as gender, age, occupation, and race, but also a higher perceived risk of getting infected with COVID-19 and previous influenza vaccination are associated with vaccine hesitancy in HCWs [16]. Nurses seem less willing to get vaccinated compared to other HCWs, such as physicians or research scientists [18,19,20,21]. Insufficient knowledge [22] or using primarily social media as information sources were also reasons for vaccine hesitancy [23]. Generally, the main reasons for hesitation were concerns about vaccine safety, efficacy, and potential side effects, at least in part due to the relatively new mechanisms of immunization [24]. We previously showed that among German HCWs, fear of short- as well as long-term vaccination side effects are also associated with vaccine hesitancy [12].

Understanding reasons for COVID-19 vaccine hesitancy is a starting point for influencing and increasing vaccination rates. Yet, so far, we do not know which potential short- and long-term side effects in particular cause HCWs to worry and possibly hesitate to get vaccinated. We therefore analyzed free-text comments on the individual fears and concerns regarding the COVID-19 vaccination side effects from that previous study.

By recognizing which side effects cause fear or concerns in HCWs, these can be addressed directly and clarified. These findings could even help define relevant outcomes in vaccine research. Thus, the aim of this paper is to gain a more nuanced view regarding the link between vaccination side effects and vaccine hesitancy in German HCWs, and to outline potential relationships between certain concerns and HCWs’ (un-)willingness to get the COVID-19 vaccine.

## 2. Materials and Methods

### 2.1. Data Collection

The data presented were collected as part of our study on vaccine acceptance and hesitancy of German health care workers [12]. This study was performed as part of research on informational needs in the CEOsys project (COVID-19 evidence ecosystem, COVID-evidenz.de). The aim of the CEOsys research group was to provide COVID-19- related evidence syntheses for different target groups, e.g., health care workers, politicians, and laity. As the CEOsys group is part of the network of German university hospitals (Netzwerk Universitätsmedizin, NUM), it was funded by the German Federal Ministry of Education and Research (Bundesministerium für Bildung und Forschung, BMBF), funding number: 01KX2021.

The original dataset was collected in a voluntary open online survey conducted from 2 February 2021 to 28 February 2021 in German language, using a cross-sectional exploratory [25] study design [12]. During the survey time, the approved vaccines Comirnaty^®^ by BioNTech/Pfizer and Vaxzevria^®^ by Astra Zeneca were in use in Germany. For prioritized groups as HCWs, vaccinations had begun approximately one month prior to the survey.

The survey link was sent to a total of 3924 email addresses of nursing homes, medical practices, ambulance services, medical universities, hospitals, ambulatory care services, and medical societies across Germany [12]. As very sensitive data (including political opinions and health issues) were collected in the original survey, strict data protection was applied. Therefore, only email addresses not containing any personally identifying attributes (such as first and/or last names) were considered. The existence of an anonymous email address was chosen as the criterion for the random selection of the facilities. Creating a stratified sample [26], all email addresses were taken from publicly accessible hospital registries, online telephone books, and online physician registries for all participant groups in each German federate state. For the sample, the addresses were taken from publicly accessible registries such as hospital registries, online telephone books, and online physician registries for each HCW group in each German state. As no openly accessible full German general medical registry exists, local/federal state online registries were searched. For doctors’ practices, the directories were searched for all available physician specialties. All anonymous email addresses of physicians’ practices, physicians’ associations and medical societies, and all medical faculties and all emergency services with anonymous email addresses provided online that we found were contacted. In non-university hospitals, the first five hospitals with an anonymous board email address were contacted in each federal state. In outpatient care services, a maximum of 50, and in nursing homes, a maximum of 40 anonymous email addresses per federal state were chosen (the first 50/40 with anonymous email addresses) [12]. The number of invitations and responses per federal state can be found Appendix A. The recipients were asked to forward the link within their institution (snowball sampling [26]). The survey was conducted using the online survey platform SoSci Survey [27]. The complete question set (54 items) can be found in our study group’s publication on HCWs’ COVID-19 vaccine acceptance and hesitancy [12] and in Appendix A. The survey was pretested by 17 members of the CEOsys network and of the authors’ departments. The complete Checklist for Reporting Results of Internet E-Surveys (CHERRIES) [28] can be found in the supplement of Holzmann-Littig et al. [12]. As in our manuscript on vaccine acceptance and hesitancy, incomplete questionnaires, questionnaires without informed consent, and questionnaires with missing information on vaccination willingness/hesitancy were excluded from statistical analyses [12].

### 2.2. Category Construction and Qualitative Analysis

The assignment of individual occupational groups to grouped categories can be found in Appendix A. If respondents replied to the questions about their professional groups or work settings with “other”, they were asked to fill in a free-text comment to indicate their profession and work setting. These free-text comments were analyzed, and participants were assigned to predefined professional groups.

The work settings were also grouped for statistical analysis in Appendix A.

The questionnaire included a question about previously experienced serious adverse vaccination effects that required medical treatment. If answered affirmatively, up to ten free-text comments about experienced symptoms could be entered.

The free-text responses were evaluated using content analysis [29]: First, within inductive categorization, fine categories were formed according to the symptoms or concerns named by participants. This category system was then critically evaluated and adjusted again. The categories were then summarized into upper categories. In individual cases, a response was assigned to more than one upper category. The symptom assignment as well as the category formation were discussed between a psychosomatic specialist (CA) and two internal medicine/nephrology physicians (CHL, CSCH), and a clinical psychologist (TF). The assignments of the categories and mentions of the participants can be taken from Appendix A.

The survey also included a question about fear of short-term side effects of the COVID-19 vaccine (“I am afraid of short-term adverse effects from COVID-19 vaccines.”). Additionally, all participants were given the option of leaving a free-text comment on short-term side effects (“If you worry about short-term side effects, which are these?”). Again, the free-text comments were evaluated by content analysis [29]. The methodical approach was the same as for the open-ended question about experienced adverse effects. The comments on feared short-term vaccination side effects were first assigned to 41 fine categories, from which the 12 upper categories were derived. Categories were discussed between CA, CHL, CSCH, and TF. These were, for example, insulting or political/non-medical statements. The assignments to the categories and mentions of the participants can be taken from Appendix A.

The survey also contained a question about fear of long-term side effects of the COVID-19 vaccine (“I am afraid of long-term adverse effects from COVID-19 vaccines.”). In addition, all participants were given the option of leaving a free-text comment on these long-term side effects (“If you worry about long-term side effects, which are these?”). Free texts on the mentioned long-term side effects of a COVID-19 vaccination were evaluated using the same inductive categorization approach as before. Category formation was discussed between CA, CHL, CSCH, and TF. See Appendix A.

### 2.3. Quantitative Statistical Analysis

Data are presented as absolute and relative frequencies, and are visualized by bar charts or in word clouds, where the text sizes are associated with the frequency of the corresponding term. For group comparisons, odds ratios with corresponding 95% confidence intervals (median-unbiased (mid-*p*) estimation provided in the R package “epitools”) are presented, and chi-square tests were performed. All tests were performed two-sided and a significance level of 5% was used. Due to the exploratory nature of the study, no adjustment for multiple testing was considered.

The free-text responses were coded in Microsoft^®^ Excel^®^, version 2013.

For the analysis on the association between attitudes towards the COVID-19 vaccination within colleagues and friends, and fear of short-term side effects and vaccination reactions were excluded in order to analyze the association with more severe expected short-term adverse effects.

Statistical analysis was performed using R, version 4.1.2 (R Foundation for statistical Computing, Vienna, Austria) and its libraries “epitools”, “arsenal”, and “wordcloud”.

### 2.4. Ethics

This study adheres to the Declaration of Helsinki. Prior to the study, approval from local ethics committees (41/21 S), data protection officers, hospital boards, and staff councils were obtained. By clicking on a checkbox at the end of the information on the study and data protection, every participant gave informed consent prior to the survey.

## 3. Results

### 3.1. Data Basis

Of the received 5448 surveys, 948 surveys were excluded for missing consent (these had been abandoned by participants at the participant information screen), not being completed or for missing information on vaccination hesitancy/willingness, therefore, 4500 questionnaires were analyzed. A detailed description of the study population is depicted in Holzmann-Littig et al. [12] Figure 1.

Overall results can be found in Appendix A.

### 3.2. Experienced Side Effects

#### 3.2.1. Frequencies of Experienced Side Effects

Results of the survey revealed that 136 participants (3.1%, 41 missing answers) stated that they had previously experienced a serious vaccination side effect and all of them named their specific side effects in the free-text comments box.

The most commonly experienced side effect was a general vaccination reaction (1.2%; *n* = 52), followed by skin damage (0.7%; *n* = 29), and allergic reaction (0.5%; *n* = 23). Relative frequencies are illustrated in the word cloud as shown in Figure 2.

#### 3.2.2. Associations between Experienced Side Effects and Fear of Side Effects after a COVID-19-Vaccination

While 136 (3.1%) participants stated to have experienced serious vaccination side effects before in free-text comments, 674 (15.0%) left free-text comments on feared short-term and 769 (17.1%) of long-term side effects. Figure 3.

Participants who stated they had experienced serious side effects with prior vaccinations that required medical treatment were more likely to fear short- as well as long-term side effects of the COVID-19 vaccination. The category construction/classification for experienced side effects can be found in Appendix A. Additionally, 45/136 participants who had previously experienced side effects (33.1%) entered free-text comments on feared short-term side effects, other/more severe than a vaccination reaction. Of those stating that they had not experienced such side effects by a prior vaccination, only 291/4323 (6.7%) feared such side effects. This difference was significant, *p* < 0.001. The category construction/classification of feared short-term side effects can be found in Appendix A. In the group stating to have previously experienced serious side effects, 37/136 (27.2%) feared a vaccination reaction; this was the case in 438/4323 participants without previously experienced adverse effects (10.1%). Additionally, this difference was significant, *p* < 0.001.

In the group of those stating to have experienced serious side effects, 53/136 (39.0%) left a free-text comment on feared long-term side effects; this was the case in only 700/4323 (16.2%) participants who had not experienced such side effects (*p* < 0.001). The category construction/classification of feared short-term side effects can be found in Appendix A.

### 3.3. Fear of Short-Term Side Effects

#### 3.3.1. Frequencies of Feared Short-Term Side Effects

Results revealed that 674/4500 respondents (15.0%) left a free-text comment on one or more particular short-term side effects they feared or worried about. Participants most often mentioned concerns regarding symptoms of a vaccination reaction (10.6%; *n* = 476). However, 344/4500 (7.6%) participants gave free-text comments on feared short-term side effects beyond a vaccination reaction. It was found that 153/4500 (3.4%) feared an allergic reaction, and 56/4500 (1.2%) feared limitations in their daily lives after the vaccination, as shown in Figure 4.

#### 3.3.2. Feared Short-Term Side Effects and Vaccine Hesitancy

Results revealed that 262/4125 (6.4%) participants who were willing to vaccinate or had vaccinated against COVID-19 left free-text comments about fears of short-term side effects beyond a vaccine reaction. For the category construction and classification, see Appendix A. Among participants who were hesitant or undecided, these were 82/375 (21.9%), *p* < 0.001.

In contrast, no significant difference was found with regards to an expected vaccination reaction as 436/4125 (10.6%) of the willing participants and 40/375 (10.7%) of the hesitant or undecided participants stated in the free-text comments that they worried about a vaccination reaction, *p* = 0.953, as shown in Appendix A.

Another significant difference was found for an expected allergic reaction where 121/4125 (2.9%) participants, who are open to get vaccinated, provided a free-text comment indicating a fear of this complication, compared to 32/375 (8.5%) for hesitant/undecided participants, *p* < 0.001. All associations between feared short-term side effects and willingness to vaccinate are shown in Figure 5 and Appendix A.

### 3.4. Fear of Long-Term Side Effects

#### 3.4.1. Frequencies of Feared Long-Term Side Effects

Results revealed that 769/4500 respondents (17.1%) left a free-text comment naming the particular long-term side effects they feared. The category construction/classification can be found in Appendix A.

Regarding long-term side effects, the most frequent reply was fear of immune/autoimmune reactions/side effects (212/4500, 4.7%). The second most common reply was the fear of neurological side effects (180/4500, 4.0%); 168/4500 (3.7%) respondents mentioned being concerned about the current unpredictability of actual long-term side effects of the COVID-19 vaccine (“unpredictable side effects”) due to the vaccine’s novelty and therefore, the lack of long-term studies. Relative frequencies are illustrated in Figure 6 and Appendix A.

#### 3.4.2. Feared Long-Term Side Effects and Vaccine Hesitancy

Strong associations between unwillingness or indecisiveness to get vaccinated and a fear of long-term side effects were found; 189/375 (50.4%) of those unwilling or undecisive left a free-text comment on feared long-term side effects. Of those willing to get vaccinated, only 580/4125 (14.1%) left such a comment, *p* < 0.001. The most common feared side-effects were immune/autoimmune reactions (unwilling/undecisive *n* = 61/375 (16.3%) vs. willing 148/4125 (3.6%), *p* < 0.001), unknown side effects (unwilling/undecided: 40/375 (10.7%) vs. willing 128/4125 (3.1%), *p* < 0.001) and neurological side effects (unwilling/indecisive: 41/375 (10.9%)) vs. willing 139/4125 (3.4%), *p* < 0.001). Further, the fear of infertility or damage to unborn children caused by the COVID-19 vaccination was associated with unwillingness/indecisiveness to get vaccinated (54/375 (14.4%) vs. willing 91/4125 (2.2%), *p* < 0.001), as shown in Figure 7 and Appendix A.

### 3.5. Associations between Professional Groups and Feared Side Effects

#### 3.5.1. Associations between Professional Groups and Fear of Short-Term Side Effects

Fear of symptoms of a vaccination reaction was most common within the non-physician medical staff (12.3%; *n* = 106/859) and the group of administrative, scientific, and other staff (12.3%; *n* = 51/414), and slightly less common among physicians (10.1%; *n* = 168/1841) and medical students (9.6%; *n* = 131/1365). In the group of physicians, the percentage of participants naming fears of short-term side effects other than vaccination reactions was higher than in other professional groups: physicians: 9.2%; *n* = 169/1841, non-physician medical staff: 7.3%; *n* = 63/859 and administration/science/other: 7.2%; *n* = 30/414, students: 78/1365, 5.7%. Bar charts are depicted in Figure 8. Working in an intensive care unit (ICU) was not associated with a significant difference in the frequency of feared short-term adverse events (ICU: 77/1525, 7.0%) vs. non-ICU 100/1105, 6.6%, *p* = 0.678). There were also no significant differences between the frequency of care for COVID-19 patients and feared short-term side effects, as shown in Appendix A.

#### 3.5.2. Associations between Professional Groups and Fear of Long-Term Side Effects

The most frequently named long-term side effects and their distributions across professional groups can be seen in Figure 8.

Overall, free-text comments on feared long-term side effects were most often provided by the group of administrative/scientific and other staff (93/414, 22.5%), followed by non-physician medical staff with 154/859 (17.9%), medical students (220/1365, 16.1%), and physicians (296/1841, 16.1%).

Immune/autoimmune reactions were more often mentioned by medical students (72/1365, 5.3%) and physicians (88/1841, 4.8%) than by administrative/scientific/other staff (18/141, 4.3%) or non-physician medical staff (30/859, 3.5%).

Both neurological side effects (23/414, 5.6%) and unpredictable side effects (26/414, 6.3%) were mentioned most often within the group of scientific/administrative and other staff.

Fear of infertility or damage to unborn children was more often cited by medical students (63/1365, 4.6%), non-physician medical staff (34/859, 4.0%), and within the group of administrative/scientific/other staff (16/414, 3.9%) than by physicians (31/1841, 1.7%), as shown in Figure 9.

Working in an ICU was not associated with a significant difference in the frequency of feared long-term adverse events (ICU: 167/1525, 15.1%) vs. non-ICU 263/1105, 17.2%, *p* = 0.144). There were also no significant differences between the frequency of care for COVID-19 patients and feared long-term side effects, as shown in Appendix A.

### 3.6. Attitude toward Vaccination within Colleagues and Friends, and Associations with Fear of Short-Term Side Effects

Strong associations were found between the fear of short-term side effects beyond a vaccination reaction and COVID-19 vaccination hesitancy in the personal environment. This effect was observed when the primary care physician had advised against a vaccination: 8/41 (19.5%) of those who were advised against the vaccination left a comment on feared short-term side effects vs. 36/518 (6.9%) who were advised to take the vaccination feared short-term side effects, *p* = 0.010. However, if colleagues did not want to get vaccinated, this effect was even stronger: 43/173 (24.9%) of those whose colleagues had decided not to get vaccinated feared short-term side effects vs. 225/3602 (6.2%) of those whose colleagues were willing to get vaccinated, *p* < 0.001. In cases where family members did not want to get vaccinated, 57/251 (22.7%) participants left a free-text comment on feared short-term side-effects vs. 223/3682 (6.1%) of the cases where family members wanted to get vaccinated, *p* < 0.001, as shown in Figure 10.

Regarding attitude toward vaccination within colleagues and friends, and associations with fear of long-term side effects, similar associations could be shown for fear of long-term side effects.

If the primary care physician had advised against a vaccination, there was an association with expected long-term adverse effects (advice against vaccination 17/41 (41.5%) vs. advice for vaccination 72/518 (13.9%), *p* < 0.001). This was also true if colleagues (advice against vaccination 78/173 (45.1%) vs. advice for vaccination 521/6302 (14.5%), *p* < 0.001) or family/friends (advice against vaccination 125/251 [49.8%] vs. advice for vaccination 503/3682 (13.7%), *p* < 0.001) had advised against a vaccination, as shown in Figure 10.

## 4. Discussion

Health care workers play an important role in increasing the anti-COVID vaccination rate and thereby containing the pandemic spread of SARS-CoV-2. They function as multipliers and role models for patients as well as for colleagues, friends, and families. However, vaccine hesitancy within HCWs has been found in numerous studies, indicating that one of the main reasons for hesitating is the fear of possible side effects [24]. In Germany, the basic immunization rate of the population is 76.1% (13 April 2022), with significant differences; for example, it is 39.3% in the group of 5–17-year-olds and 88.8% in those over 60. Still, HCWs continue to represent being important ambassadors of the COVID-19 vaccination for the non-vaccinated, but also for the booster vaccination (here, the rate is 59.0%) [30]. As HCWs serve as advisors on vaccination to their patients, it remains important to address fears of side effects so as not to jeopardize the gains achieved, especially given that recent studies indicate that due to new virus variants, up to 90% of the population need to be vaccinated against COVID-19 in order to achieve herd immunity [4].

The aim of this paper was to further differentiate which side effects of the anti-COVID-vaccination HCWs are concerned with, and which might be associated with unwillingness or indecisiveness to get vaccinated.

### 4.1. Experienced Side Effects

Only a relatively small number of participants reported having experienced adverse effects after a previous vaccination. The most frequently mentioned vaccination reactions were fever, local pain, dizziness or an allergic reaction. This is in line with the literature, where 1.8–14.4 cases of anaphylactic reactions [31,32] and 1 immune thrombocytopenic purpura per 40,000 vaccinated children [33] have been reported.

In our study, participants, who reported previously experienced serious side effects after a vaccination that required medical treatment, were more likely to fear short- as well as long-term side effects of the COVID-19 vaccination. This seems conclusive as people who have already experienced serious side effects might worry more about a repetition of this experience. In line with this, Rzymski and colleagues [34] described that side effects experienced in the context of previous anti-COVID vaccinations were associated with a refusal of a booster vaccination. In addition to that, we previously demonstrated that participants who had already experienced a serious side effect were more hesitant [12]. To address these concerns, these particular individuals could receive an offer to be vaccinated in specially equipped centers or hospitals in order to be able to cater for the event of a possible anaphylactic reaction.

However, the overall number of participants, who had experienced such side effects, was significantly lower than the number of those who feared side effects. Thus, having experienced side effects does not seem to sufficiently explain the concerns.

### 4.2. Fear of Short- and Long-Term Side Effects

When asked about short-term side effects, many participants worried about symptoms that could be summarized under general vaccination reactions, such as exhaustion, fever, dizziness, and headache. This indicates that many HCWs worry about side effects that are common and likely to occur after an anti-COVID vaccination [35]. Moreover, allergic reactions as well as possible limitations to their daily functioning were another concern. More severe adverse effects, such as cardiopulmonary symptoms, organ damage or death were only mentioned rarely.

Long-term side effects were stated slightly more often. However, HCWs mostly cited long-term side effects such as neurological or immune and autoimmune symptoms or diseases. Again, these constitute adverse effects that could actually occur after an anti-COVID vaccine, even though the probabilities remain rather low, and there is a substantially higher risk of these neurological outcomes after a SARS-CoV-2 infection [36].

It is noteworthy that one of the most commonly reported fears was not knowing which possible adverse reactions might emerge in the future. Since mRNA-vaccinations are still relatively new and innovative, data on their long-term effects are not yet available. At the time our survey was conducted, only the vaccines Comirnaty^®^ by BioNTech/Pfizer and Vaxzevria^®^ by AstraZeneca had been approved and were in use in Germany.

The initial fear caused by the lack of (long-term) data on these new vaccines might have declined within the first year of applying these vaccines, and with research showing that they very rarely cause serious side effects [37] whilst protecting from severe courses of COVID-19 [38,39].

### 4.3. Fear of Side Effects and Association with Vaccine Hesitancy

Fear of symptoms of a vaccination reaction (i.e., local reaction, fever, drowsiness, etc.) were distributed equally between those willing, unwilling or undecided about receiving an anti-COVID vaccination. Thus, apprehension of such symptoms, which are likely to occur after vaccinations in general [32]), and are also common after COVID-19 vaccinations [40,41], does not seem to be a reason for refusal of vaccination in general.

On the other hand, fear of more serious short-term side effects was mentioned significantly more often in the group of hesitant or unwilling participants. It therefore seems necessary to provide sufficient information about short-term side effects and their likeliness of occurrence as well as their severity.

Nearly half of the participants who indicated they declined or were undecided about vaccination had left comments about feared long-term side effects. Especially, autoimmune-/immune reactions were frequently named. However, such reactions occur only in very rare individual cases [42]. Targeted education should be offered here. Fear of neurologic side effects was also frequently mentioned, which are rare, too [42]. Additionally, many participants stated that they feared unpredictable side effects. This concern could only be mitigated to a limited extent, especially at the beginning of the vaccination campaign since the vaccines had not been in use for long. Although, experience with the mode of action of these vaccines did of course exist. Today, one year later, it is precisely this concern that can be refuted by the vast amount of data that has been collected from vaccinating millions of people worldwide. This should be considered in vaccination campaigns targeting the hesitant part of the population. Moreover, the fear of negative effects on fertility or on unborn children, stated by some participants, are common arguments against the COVID-19 vaccination, although there is no evidence of such side effects [43].

### 4.4. Associations between Professional Groups and Fear of Side Effects

Previous findings suggested an association between profession and vaccine hesitancy, with some professional groups being more hesitant than others [18,21,22]. Our results are in line with this. Physicians seem to worry less about symptoms of vaccination reaction, but they indicated a bigger fear of other short-term side effects such as allergic reactions or limitations in daily lives.

Interestingly, physicians named feared long-term side effects less often than all other groups while short-term side effects had been named most often by this group. Fears of individual side effects differed between the occupational groups studied. This may suggest that education on COVID-19 vaccination should be tailored to address side effects that are frequently feared in the particular group. Furthermore, the knowledge of how the vaccinations work, which is potentially unequal in the different professional groups, should be supplemented. Further studies should evaluate the reasons for these differences in order to provide group-specific education on COVID-19 vaccination.

### 4.5. Attitude toward Vaccination within Colleagues and Friends, and Associations with Fear of Side Effects

There was a strong association between feared short- and long-term side effects, and the general practitioner’s advice not to be vaccinated. This seems conclusive since it can be assumed that the general practitioner’s expectation of side effects may have led to his/her advice. Conversely, it was also shown that when participants had advised their patients or families/acquaintances against the anti-COVID vaccination, there was a strong association with feared side effects of vaccination. This shows all the more clearly what an ambassadorial role health care workers have, and that good and continuous education of all HCW professional groups is essential.

At the same time, there was a strong association between existing concerns of short- and long-term side effects, and the fact that colleagues or family members/acquaintances had advised the participant against vaccination. Although no causal relationship can be derived from this, it could be an indication of the formation of clusters in which a rejection of vaccinations and a fear of particular side effects might manifest. Such clusters could also arise not only in families and between acquaintances, but also in the work environment. Although the willingness to vaccinate and the vaccination rates among health care workers in Germany are high [11,12,13], it seems conceivable that such clustering could lead to significantly lower vaccination rates among staff in individual areas, with all the resulting dangers. The fear of symptoms triggered by vaccination might facilitate emotional processing [44], as well as lead to false beliefs about adverse effects and possible risks of the anti-COVID vaccine. By teaming up with others, who share these concerns and are hesitant about vaccination, possible in-group favoritism can occur [45]. Both emotional processing and in-group favoritism can enable confirmatory information search or selective exposure to information that supports pre-existing views or beliefs [46]. Therefore, such clusters would continually reinforce each other’s opinions and possibly reject information that could weaken or disprove their convictions. It seems doubtful whether such groups can be reached through general campaigns. It should be further investigated whether direct contact, e.g., within seminars and training courses, can be helpful here.

### 4.6. Strengths and Limitations

This work has several strengths. First, it is one of the world’s largest vaccination readiness surveys among health care professionals. Secondly, due to the numerous free-text comments, we were able to carry out a well-founded analysis of the nature of fears of side effects, and show the associations of these fears with professional groups and rejection of the vaccination. This can help design further vaccination campaigns that are even better adapted to the target group.

Some weaknesses have to be mentioned. The online survey might be prone to selection bias. However, especially during the pandemic and due to the short time available, a pen and paper survey was not reasonable. In addition, the method of snowball sampling was used, so that no response rate could be calculated. Yet, participants from all parts of Germany and all areas of health care could be reached [12]. In addition, a self-selection bias among more positive participants cannot be ruled out with absolute certainty. However, the invitation text was worded neutrally, and the nature of some negative comments from participants shows that the survey may also have been used to express dissatisfaction with the vaccination. It should also be taken into account that at the time of the survey, only the vaccines Vaxzevria^®^ from AstraZeneca and Comirnaty^®^ from Biontech/Pfizer were available in Germany [47,48], and that the vaccination campaign had only started about a month and a half before the beginning of the survey. In the following months, case reports on vaccination side effects were more widely communicated in the press. There were also changing recommendations of the Standing Commission on Vaccination regarding different vaccines [49]. This may have exacerbated the fears depicted here. Therefore, follow-up studies on these concerns appear to be necessary, and a good education for HCWs about the general occurrence and frequency of side effects according to current knowledge is all the more urgently needed.

## 5. Conclusions

In our work, fear of vaccination reactions was not associated with vaccination refusal. However, fears of serious short- and long-term side effects were very clearly associated with vaccination rejection. Our data may help address fears of adverse effects of COVID-19 vaccination to facilitate rational and well-informed decision making by the respective target groups. Thorough and transparent information about the frequency and nature of expected vaccination side effects is urgently needed, especially for health care professionals. This also includes transparent communication of any side effects that may occur. Furthermore, it was shown that especially when the circle of acquaintances of the participants advises against vaccination, they are significantly more likely to worry about side effects (short- and long-term). Thus, further education for medical staff seems to be necessary in order to achieve good information transfer in these clusters. However, this may not be achieved with general poster campaigns or impersonal campaigns alone. More personal approaches, such as seminars and webinars, also with the possibility of asking follow-up questions, could be a useful addition. Educating about side effects and thereby reducing the fear of these is important, especially when considering how (mis)information can be transferred within groups of colleagues or friends/families as well as between HCWs and patients. Considering possible expansion of the new vaccine production or the new mode of action, information campaigns on tolerability are still necessary after the COVID-19 pandemic has subsided.

## Figures and Tables

**Figure 1 vaccines-10-00689-f001:**
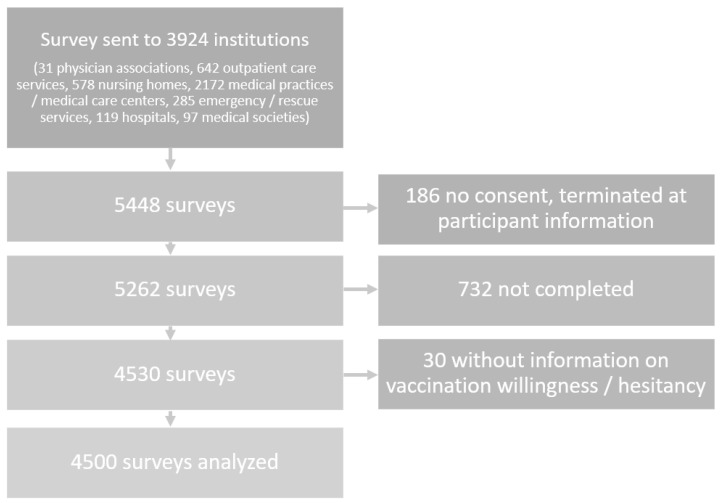
Flow Chart, adapted from Holzmann-Littig et al. [12].

**Figure 2 vaccines-10-00689-f002:**
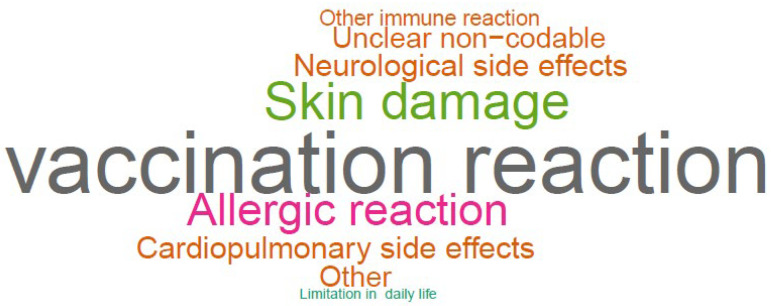
Word Cloud of experienced side effects, constructed from upper categories.

**Figure 3 vaccines-10-00689-f003:**
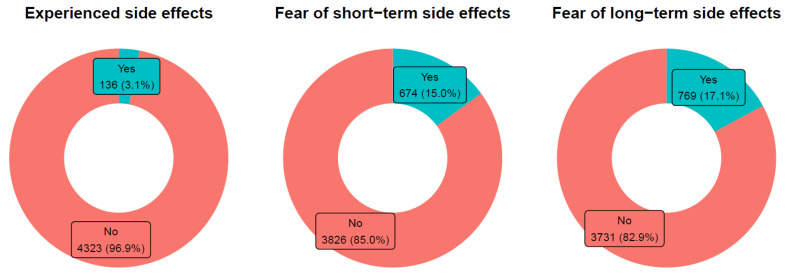
Frequencies of participants with experienced or feared side effects.

**Figure 4 vaccines-10-00689-f004:**
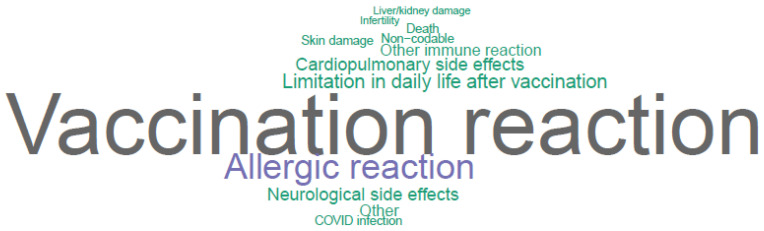
Word cloud of feared short-term side effects, constructed from upper categories.

**Figure 5 vaccines-10-00689-f005:**
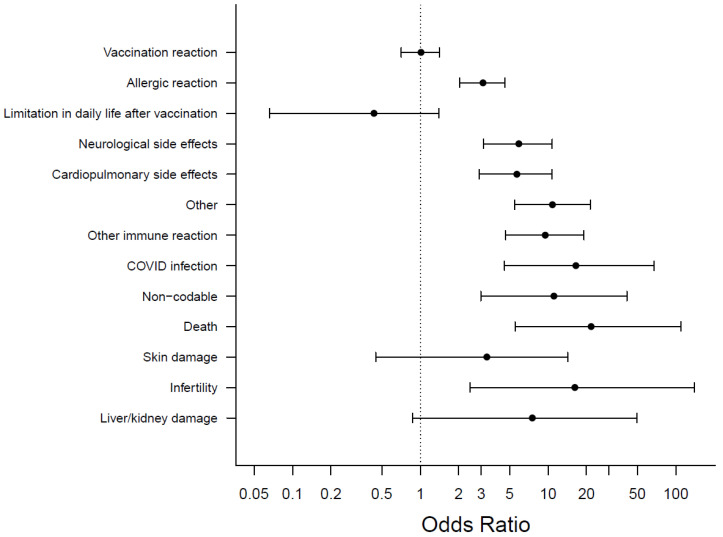
Feared short-term side effects and Odds Ratios for COVID-19 vaccination hesitancy, with categories arranged by frequency, most frequent on top.

**Figure 6 vaccines-10-00689-f006:**
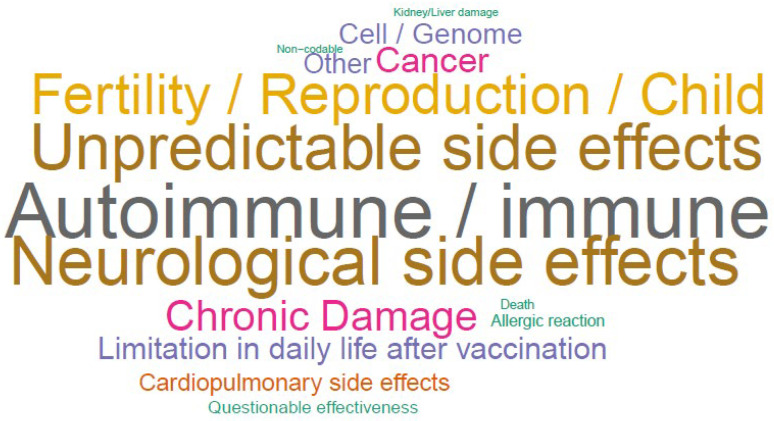
Word cloud of feared long-term side effects, constructed from upper categories.

**Figure 7 vaccines-10-00689-f007:**
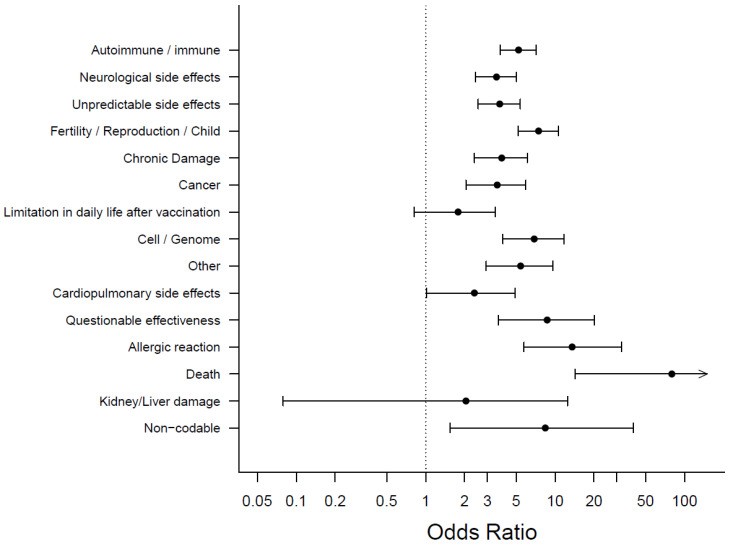
Feared long-term side effects and Odds Ratios for COVID-19 vaccination hesitancy, with categories arranged by frequency, most frequent on top. Upper limit of the 95% confidence interval for “death” is 2006.

**Figure 8 vaccines-10-00689-f008:**
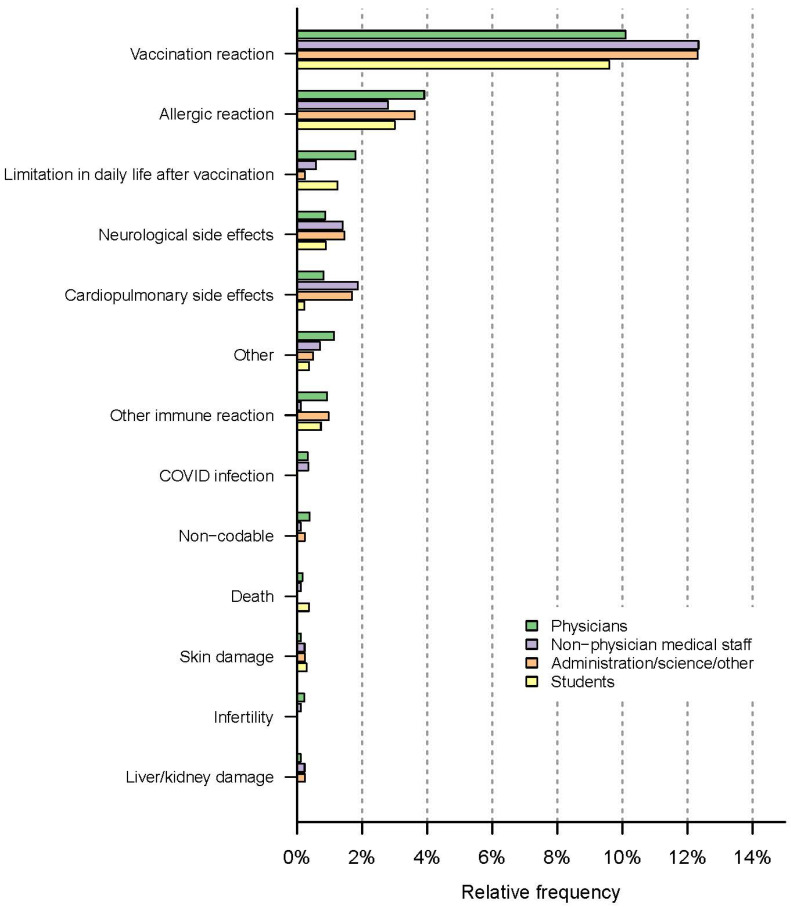
Professional groups and feared short-term side effects.

**Figure 9 vaccines-10-00689-f009:**
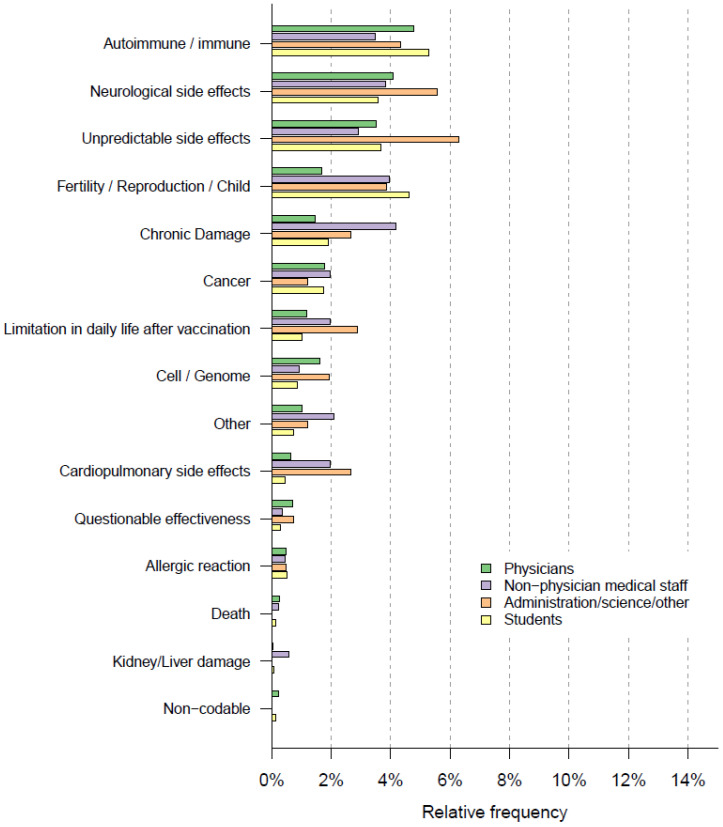
Professional groups and feared long-term side effects.

**Figure 10 vaccines-10-00689-f010:**
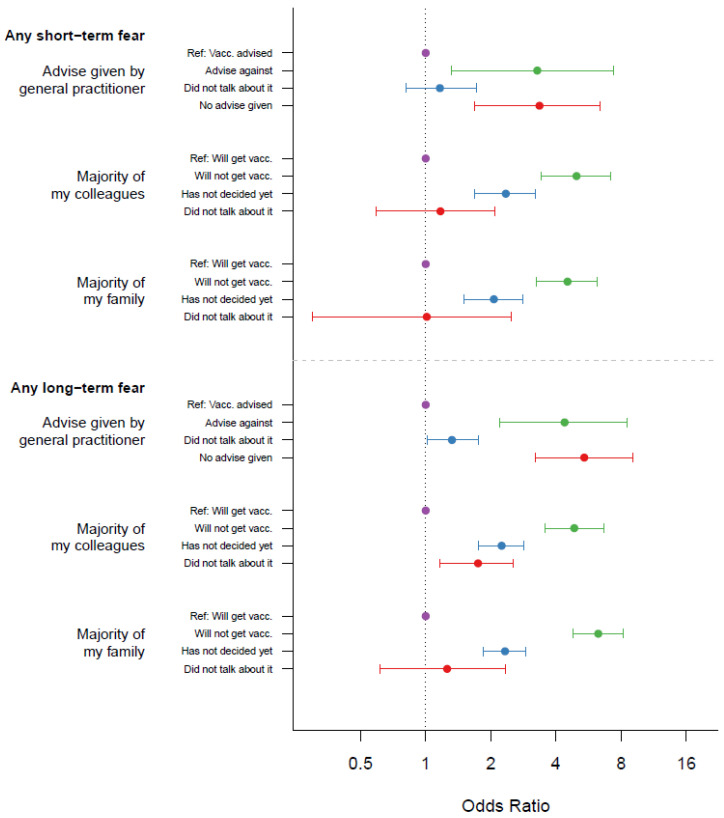
Attitudes towards the COVID-19 vaccination in the personal environment and Odds ratios for side effects (short-term: vaccination reactions excluded).

## Data Availability

The datasets for this manuscript are not publicly available because written informed consent excluded data sharing, as advised by the local data protection officer in accordance with the German data protection law.

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
