# Peer review of "COVID-19 Vaccines: Fear of Side Effects among German Health Care Workers"

_vaccines, 2022, doi:10.3390/vaccines10050689_

Round 1

Reviewer 1 Report

Holzmann-Littig and colleagues proposed an interesting research article on the fear perception of healthcare workers about COVID-19 vaccination in Germany. In particular, the authors analyzed the data obtained from an online survey proposed to German HCWs. The authors collected a huge number of participants collecting free-text answers to several questions. Overall, the manuscript is interesting, however, there are some minor/major comments that the authors have to address before publication. Please, see the comments below:
1) In the Introduction section, the authors should briefly describe the vaccination rates reached in Germany among the general population and HCWs. This information is fundamental to understand if a real issue related to people's adherence to vaccination exists. Please clarify this point;
2) Although the authors state: “The survey was conducted using the online survey platform SoSci Survey (25). The complete question set (54 items) can be found in our study group’s publication on HCW’s COVID-19 vaccine acceptance and hesitancy (22).”, the authors are encouraged in reporting the questionnaire adopted as supplementary material;
3) What did the authors mean with “4500 questionnaires could be analyzed.”. Is this the exact number of questionnaires analyzed?
4) Check the grammar of “3.3.1 Frequencies of feares of short-term side effects”;
5) Did the authors stratify HCWs according to their ward? In particular, it would be interesting to evaluate the fear perception in HCWs working in the ICU or infectious diseases wards compared to the data obtained in HCWs working in other wards like cardiology, gastroenterology, etc.

Author Response

Reviewer 1:
We thank the reviewer for his/her valuable comments and have done
our utmost to answer them to his/her full satisfaction. If the reviewer has
any questions, we are of course very happy to answer them at any time.
Our answers are listed below point by point.
Comment 1: Holzmann-Littig and colleagues proposed an interesting
research article on the fear perception of healthcare workers about
COVID-19 vaccination in Germany. In particular, the authors analyzed
the data obtained from an online survey proposed to German HCWs.
The authors collected a huge number of participants collecting free-text
answers to several questions. Overall, the manuscript is interesting,
however, there are some minor/major comments that the authors have
to address before publication.
Response 1: We thank the reviewer for this encouraging feedback and
are pleased to address the points below.
Comment 2: In the Introduction section, the authors should briefly
describe the vaccination rates reached in Germany among the general
population and HCWs. This information is fundamental to understand if
a real issue related to people's adherence to vaccination exists. Please
clarify this point
Response 2: We thank the reviewer for this valuable comment.
Fortunately, vaccination rates of up to 92% have now been achieved in
hospitals in Germany. However, especially due to the presumed
clustering, significantly lower vaccination rates with the resulting
dangers may occur in individual areas. Here it is still important to be
able to address the fears of these individuals well. We also believe that
the data can potentially support vaccination campaigns in other states
where these rates have not yet been achieved. We have added the
following sentences to our introduction: “In Germany, according to a
study by the Robert Koch Institute, the vaccination rate of German
hospital staff was 83% (primary vaccination) in April 2021, with
significant differences physicians and non-physicians (11). Our study
had already suggested a COVID-19 vaccination acceptance rate of
91.7% of healthcare professionals in Germany in 2021 (12). This figure
is mirrored at least for German hospital staff according to a recent study
by the Robert Koch Institute, with a rate of 92% fully vaccinated study
participants (13). However, there is still a proportion of unvaccinated
health care workers in Germany.” (p. 2, ll. 71-78) The following sentence
has been added to the discussion: “Although the willingness to
vaccinate and the vaccination rates among healthcare workers in
Germany are high (11-13), it seems conceivable that such clustering
could lead to significantly lower vaccination rates among staff in
individual areas, with all the resulting dangers.” (p. 17, ll. 495-497)
Comment 3: Although the authors state: “The survey was conducted
using the online survey platform SoSci Survey (25). The complete
question set (54 items) can be found in our study group’s publication on
HCW’s COVID-19 vaccine acceptance and hesitancy (22).”, the authors
are encouraged in reporting the questionnaire adopted as
supplementary material
Response 3: We thank the reviewer for this comment and agree that it
will make it easier for readers to find the survey questions directly in the
supplement of this manuscript. We have added this to the Supplement.
Comment 4: What did the authors mean with “4500 questionnaires
could be analyzed.”. Is this the exact number of questionnaires
analyzed?
Response 4: We thank the reviewer for pointing out that this sentence
was obviously ambiguous. It is the exact number of questionnaires that
have been analyzed. We have changed the sentence as follows: “4500
questionnaires were analyzed.” (p. 5, ll. 217-218)
Comment 5: Check the grammar of “3.3.1 Frequencies of feares of
short-term side effects”
Response 5: We thank the reviewer for pointing this out and have been
very happy to correct the linguistic inaccuracy. We have supplemented
the sentence as follows: “3.3.1. Frequencies of feared short-term side
effects”. (p. 7, l. 258). Similarly, we have corrected the heading for
long-term side effects: “3.4.1. Frequencies of feared long-term side
effects” (p. 8, l. 286)
Comment 6: Did the authors stratify HCWs according to their ward? In
particular, it would be interesting to evaluate the fear perception in
HCWs working in the ICU or infectious diseases wards compared to the
data obtained in HCWs working in other wards like cardiology,
gastroenterology, etc.
Response 6: We thank the reviewer for this important tip. For reasons
of data protection (avoidance of identification of individual study
participants), we did not request the ward's field of expertise. However,
first, we queried whether participants worked in an intensive care unit
and how often they worked with COIVD-19 patients. For both, no
significant differences were found with regard to feared short- or longterm
side effects. We have added the following sentences: “Working in
an intensive care unit ICU was not associated with a significant
difference in the frequency of feared short-term adverse events (ICU:
77/1525, 7.0%) vs non-ICU 100/1105, 6.6%, p=0.678). There were also
no significant differences between the frequency of care for COVID-19
patients and feared short-term side effects.” (p. 10, ll. 324-328);
“Working in an ICU was not associated with a significant difference in
the frequency of feared long-term adverse events (ICU: 167/1525,
15.1%) vs non-ICU 263/1105, 17.2%, p=0.144). There were also no
significant differences between the frequency of care for COVID-19
patients and feared long-term side effects. Supplemental Table S7.” (p.
12, ll. 348-351); In addition, we have added the data to Supplemental
Table S7.

Reviewer 2 Report

 The coronavirus disease 2019 (COVID-19) pandemic has caused and still causes impairment and detriment on health care systems, economies and foremost on the lives of millions of people worldwide. The possibility to vaccinate presents the most efficient solution in controlling and ending the pandemic.Healthcare workers (HCWs) play a key role in increasing anti-COVIDvaccination rates. Fear of potential side effects is one of the main reasons for vaccine hesitancy. By investigating the side effects of the vaccine that German medical staff were concerned about, this study analyzed the factors affecting the vaccine promotion, and provided help for the subsequent vaccine promotion. While there still have some limitations in this research which should be addressed and discussed.

Q1: By March 30, 2022, 75% of Germany's population was vaccinated against COVID-19 (https://ourworldindata.org/covid-vaccinations). COVID-19 vaccine is now widely recognized. Please explain the practical significance of this research conclusion.

Q2: Please explain how 3924 email addresses were selected. What are the reasons for choosing?

Q3: It is recommended to organize the classification description into tables or figure. For example, lines 216 to 225, 240 to 251.

Q4: Please add the arrow meaning in Figure 7.

Q5: Please check the 95%CI of "Majority of my family -Did not talk about it" in Figure 10.

Author Response

We thank the reviewer for the helpful and valuable comments and have
done our best to respond to his/her full satisfaction. If there are any
questions, we will be very happy to answer them at any time. Please
find our answers listed below point by point.
Comment 1: The coronavirus disease 2019 (COVID-19) pandemic has
caused and still causes impairment and detriment on health care
systems, economies and foremost on the lives of millions of people
worldwide. The possibility to vaccinate presents the most efficient
solution in controlling and ending the pandemic. Healthcare workers
(HCWs) play a key role in increasing anti-COVID vaccination rates.
Fear of potential side effects is one of the main reasons for vaccine
hesitancy. By investigating the side effects of the vaccine that German
medical staff were concerned about, this study analyzed the factors
affecting the vaccine promotion, and provided help for the subsequent
vaccine promotion. While there still have some limitations in this
research which should be addressed and discussed.
Response 1: We thank the reviewer for the important comments and
are very pleased to respond to them below.
Comment 2: By March 30, 2022, 75% of Germany's population was
vaccinated against COVID-19 (https://ourworldindata.org/covidvaccinations).
COVID-19 vaccine is now widely recognized. Please
explain the practical significance of this research conclusion.
Response 2: We thank the reviewer for this important comment and
would like to explain the significance more fully. We have included the
following paragraph in the discussion: “In Germany, the basic
immunization rate of the population is 76.1% (04/13/2022), with
significant differences, for example, it is 39.3% in the group of 5–17-
year-olds and 88.8% in those over 60. Still, HCW continue to represent
important ambassadors of the COVID-19 vaccination for the nonvaccinated,
but also for the booster vaccination (here the rate is 59.0%)
(30). As HCWs serve as advisors on vaccination to their patients, it
remains important to address fears of side effects so as not to
jeopardize the gains achieved, especially given that recent studies
indicate that due to new virus variants up to 90% of the population need
to be vaccinated against COVID-19 in order to achieve herd immunity
(4).” (p. 15, ll. 385-393) In addition, we have added the following
sentences to our introduction: “In Germany, according to a study by the
Robert Koch Institute, the vaccination rate of German hospital staff was
83% (primary vaccination) in April 2021, with significant differences
physicians and non-physicians.(1) Our study had already suggested a
COVID-19 vaccination acceptance rate of 91.7% of healthcare
professionals in Germany in 2021.(2) This figure is mirrored at least for
German hospital staff according to a recent study by the Robert Koch
Institute, with a rate of 92% fully vaccinated study participants.(3)
However, there is still a proportion of unvaccinated health care workers
in Germany.” (p. 2, ll. 71-78) The following sentence has been added to
the discussion: “Although the willingness to vaccinate and the
vaccination rates among healthcare workers in Germany are high (11-
13), it seems conceivable that such clustering could lead to significantly
lower vaccination rates among staff in individual areas, with all the
resulting dangers.” (p. 17, ll. 495-498)
Comment 3: Please explain how 3924 email addresses were selected.
What are the reasons for choosing?
Response 3: We thank the reviewer for this reference. We had
described this in detail in the first publication on the study and therefore
refer to the first publication as a source in this manuscript. However, we
fully agree that it is clearer for readers to read the more detailed
information in our manuscript as well. We have therefore, on the one
hand, added a table on the number of invitations per state (taken from
the first publication in 2021) to the supplement and, on the other hand,
supplemented the text in the material/methods section as follows:
“Therefore, only email addresses not containing any personally
identifying attributes (such as first and/or last names) were considered.
The existence of an anonymous email address was chosen as the
criterion for the random selection of the facilities. Creating a stratified
sample (24), all email addresses were taken from publicly accessible
hospital registries, online telephone books and online physician
registries for all participant groups in each German federate state. For
the stratified sample (4), the addresses were taken from publicly
accessible registries such as hospital registries, online telephone books
and online physician registries for each HCW group in each German
state . Because no openly accessible full German general medical
registry exists, local/federal state online registries were searched. For
doctor’s practices, the directories were searched for all available
physician specialties. All anonymous email addresses of physicians’
practices, physicians’ associations and medical societies and all
medical faculties and all emergency services with anonymous email
addresses provided online that we found were contacted. In nonuniversity
hospitals, the first five hospitals with an anonymous board
email address were contacted in each federal state. In outpatient care
services, a maximum of 50 and in nursing homes a maximum of 40
anonymous email addresses per federal state were chosen (the first
50/40 with anonymous email addresses).(2) The number of invitations
and responses per federal state can be found in Supplemental Table
S1.” (p. 3, ll. 125-142)
Comment 4: It is recommended to organize the classification
description into tables or figure. For example, lines 216 to 225, 240 to
251.
Response 4: We thank the reviewer for this important comment, we
agree that this was not completely clear. We have now added the
following sentences in the paragraph: “The category construction /
classification for experienced side effects can be found in Supplemental
Table S4.” (p. 6, ll. 242-243). “45/136 participants who had previously
experienced side effects (33.1%) entered free text comments on feared
short-term side effects any other / more severe than a vaccination
reaction.” (pp. 6-7, ll. 243-245). “The category construction /
classification of feared short-term side effects can be found in
Supplemental Table S5.” (p. 7, ll. 247-248) “The category construction /
classification of feared short-term side effects can be found in
Supplemental Table S6.” (p. 7, ll. 254-256). “For the category
construction and classification see Supplemental Table S5.” (p. 7, ll.
270-271). “ Also, we have added references to Supplemental Table S7.
We hope to provide more clarity by referring to these tables. In addition,
we have now also added the data included in the text in the section
"Associations between experienced side effects and fear of side effects
after a COVID-19-vaccination" to Supplemental Table S7.
Comment 5: Please add the arrow meaning in Figure 7.
Response 5: We agree with the reviewer that this was not completely
clear. We have added the sentence "Upper limit of the 95% confidence
interval for "death" is 2006." to the caption. (p. 10, ll. 313-314)
Comment 6: Please check the 95%CI of "Majority of my family -Did not
talk about it" in Figure 10.
Response 6: We thank the reviewer for this important comment. We
used the median-unbiased estimation (mid-p) method for calculation of
odds ratios and corresponding confidence intervals which is the default
method of the oddsratio function provided in the R package epitools.
When the absolute cell frequencies are small, this method can provide
asymmetric confidence intervals.
Only 67 particpants replied that they "Did not talk about it" with the
"Majority of my family". Only 4 of those (6.0%) stated that they fear any
short term side effects. For clarification, we added this information to
the statistical methods section: "For group comparisons, odds ratios
with corresponding 95% confidence intervals (median-unbiased (mid-p)
estimation provided in the R package "epitools") are presented, and chisquare
tests were performed." (pp. 4-5,ll. 195-197)

Round 2

Reviewer 1 Report

The authors well-addressed all of my previous comments. The manuscript can be accepted for publication after the editorial check.